# Effects of *Helicobacter pylori* and Nitrate-Reducing Bacteria Coculture on Cells

**DOI:** 10.3390/microorganisms10122495

**Published:** 2022-12-16

**Authors:** Hinako Ojima, Sakiko Kuraoka, Shyoutarou Okanoue, Hiroyuki Okada, Kazuyoshi Gotoh, Osamu Matsushita, Akari Watanabe, Kenji Yokota

**Affiliations:** 1Department of Bacteriology, Academic Field of Health Sciences, Okayama University, Okayama 700-8558, Japan; 2Department of Gastroenterology and Hepatology, Academic Field of Medicine Dentistry and Pharmaceutical Sciences, Okayama University, Okayama 700-8558, Japan; 3Himeji Red Cross Hospital, Himeji 670-8540, Japan; 4Department of Bacteriology, Academic Field of Medicine Dentistry and Pharmaceutical Sciences, Okayama University, Okayama 700-8558, Japan; 5Department of Oral Health Care and Rehabilitation, Institute of Biomedical Sciences, Graduate School, Tokushima University, Tokushima 770-0042, Japan

**Keywords:** *Helicobacter pylori*, nitrate-reducing bacteria, IL-8, TNF-α, cell cycle

## Abstract

*Helicobacter pylori* infection is an important risk factor for developing gastric cancer. However, only a few *H. pylori*-infected people develop gastric cancer. Thus, other risk factors aside from *H. pylori* infection may be involved in gastric cancer development. This study aimed to investigate whether the nitrate-reducing bacteria isolated from patients with atrophic gastritis caused by *H. pylori* infection are risk factors for developing atrophic gastritis and gastric neoplasia. Nitrate-reducing bacteria were isolated from patients with atrophic gastritis caused by *H. pylori* infection. Among the isolated bacteria, *Actinomyces oris*, *Actinomyces odontolyticus*, *Rothia dentocariosa*, and *Rothia mucilaginosa* were used in the subsequent experiments. Cytokine inducibility was evaluated in monocytic cells, and mitogen-activated protein kinase (MAPK) activity and cell cycle were assessed in the gastric epithelial cells. The cytotoxicities and neutrophil-inducing abilities of the *Actinomyces* and *Rothia* species were enhanced when cocultured with *H. pylori*. Th1/Th2-related cytokines were also expressed, but their expression levels differed depending on the bacterial species. Moreover, *H. pylori* and *Actinomyces* activated MAPK (ERK and p38) and affected cell cycle progression. Some nitrate-reducing bacteria cocultured with *H. pylori* may promote inflammation and atrophy by inducing cytokine production. In addition, the MAPK activation and cell cycle progression caused by these bacteria can contribute to gastric cancer development.

## 1. Introduction

*Helicobacter pylori* is a Gram-negative, spiral-shaped bacterium that inhabits the gastric mucosa. Approximately half of the world’s population is infected with *H. pylori* [1]. *H. pylori* infection causes inflammation of the gastric mucosa, leading to the development of gastrointestinal diseases, such as gastritis, peptic ulcer, gastric MALT lymphoma, and gastric cancer [2,3,4]. Approximately 75% of gastric cancer cases are caused by *H. pylori* infection [5]. However, only 1–2% of people infected with *H. pylori* develop gastric cancer [6]. Therefore, other risk factors may be involved in gastric cancer development.

Long-term *H. pylori* infection causes atrophy of the gastric mucosa and decreases gastric acid secretion [7]. As a result, the pH of the stomach increases, and bacteria other than *H. pylori* proliferate and colonize the low-acid gastric mucosa [8]. Recent studies on the gut microbiota have found that the presence of *H. pylori* and its symbiotic bacteria contributes to gastric cancer development [9,10]. Therefore, we focused on nitrate-reducing bacteria among the symbiotic bacteria as risk factors for developing gastric cancer. Nitrate-reducing bacteria convert nitrate to nitrite. Under conditions of hypoacidity with gastric atrophy, nitrate-reducing bacteria proliferate, thereby increasing the nitrite levels in the stomach. Nitrite produced in the human body reacts with other organic nitrogen compounds and participates in the formation of carcinogenic nitroso compounds [11,12]. This result suggests that nitrate-reducing bacteria are involved in the development of gastric epithelial cell neoplasia. However, the effects of the proliferation of nitrate-reducing bacteria on epithelial cells remain unclear.

Normal cells are affected by exogenous stimuli (e.g., ultraviolet rays and chemical substances), endogenous stimuli (e.g., metabolites), and DNA damage caused by genetic factors, which cause abnormal cell signal transduction. Consequently, cancer occurs at the cellular level [13,14]. *H. pylori* activates various cell-signaling pathways, including the mitogen-activated protein kinase (MAPK) signaling pathway [15,16], which regulates gene expression, cell proliferation, cell cycle progression, and apoptosis [17].

In this study, we aimed to investigate whether *H. pylori* and nitrate-reducing bacteria are risk factors for developing gastric cancer by evaluating their effects on inflammatory and cytotoxic cytokine production using THP monocyte cells. In addition, to examine the role of epithelial cell neoplasms, we used AGS cells to observe MAPK activation and cell cycle progression.

## 2. Materials and Methods

### 2.1. Participants

Gastric mucosal biopsy samples were collected from the gastric antrum and body of patients who were suggested to have gastritis due to *H. pylori* infection at Okayama University Hospital and underwent upper gastrointestinal endoscopy. Twenty six strains of nitrate-reducing bacteria were isolated from 79 patients with gastritis. The study was approved by the Clinical Research Ethics Committee No. K1903-025 of Okayama University. 

### 2.2. Isolation and Identification of Nitrate-Reducing Bacteria

#### 2.2.1. Isolation

The collected biopsy samples were homogenized and cultured on brain heart infusion (BHI) agar containing 7% horse-defibrinated blood. They were cultured at 37 °C under microaerophilic conditions for 1 week. Bacteria other than *H. pylori* were isolated on BHI agar for 1 week.

#### 2.2.2. Nitrate Reduction Test

Bacteria grown on BHI agar were cultured in a broth medium containing potassium nitrate for 3 days and then added with 50 µL of α-naphthylamine acetic acid solution and 50 µL of sulfanilic acid solution. A positive change in the nitrate reduction test was observed when the bouillon turned from yellow to red.

#### 2.2.3. Bacteria Identification

Bacterial DNA was extracted using the Wizard ^®^ Genomic DNA Purification Kit (Promega, Madison, WI, USA) in accordance with the manufacturer’s protocol. The 16SrDNA sequence was PCR amplified from the extracted DNA by using the following primers: 16S-F, 5′-TTGGAGAGTTTGATCCTGGCTC-3′ and 16S-R, 5′-ACGTCATCCCCACCTTCCTC-3′. The PCR reaction solution was prepared using TaKaRa Ex Taq DNA Polymerase (Takara Bio Inc., Kusatsu, Japan) in accordance with the manufacturer’s instructions. The PCR cycling program was as follows: denaturation at 94 °C for 5 min and 34 cycles of 94 °C for 10 s, 56 °C for 10 s, and 72 °C for 15 s. The PCR products were electrophoresed on a 2% agarose gel and stained with ethidium bromide. After staining, the gel was photographed using a GelScene GS-GU gel photographing device (Astec, Kasuyagunsimemachi, Japan). The PCR-amplified DNA samples were purified using ethanol precipitation. The sequences were outsourced to Hokkaido System Science Co., Ltd. (Sapporo, Japan) The sequencing data were compared with the genome analysis data using BLAST^®^ to identify the bacterial species. 

### 2.3. Bacterial Species

Table 1 lists the isolated and identified nitrate-reducing bacteria, from which the most frequently isolated indigenous oral cavity bacteria (*Actinomyces oris*, *Actinomyces odontolyticus*, *Rothia dentocariosa*, and *Rothia mucilaginosa*) were used for further analyses. These bacteria and *H. pylori* (ATCC43504) were cultured on agar. Colonies were scraped with a sterile cotton swab and suspended in physiological saline. The number of bacteria was counted at an optical density at 600 nm (OD_600_) = 1.0. 

### 2.4. Cytokine Measurement

Human monocytic THP cells were stimulated with each of the four strains of nitrate-reducing bacteria and *H. pylori*, and the cytokines in the culture supernatants and the mRNA transcription levels of the cytokines in the cells were measured.

#### 2.4.1. Sample Preparation

THP cells (10⁶ cells/mL) were stimulated with *H. pylori* (10⁶ CFU/mL), *A. oris* (5 × 10⁵ CFU/mL), *A. odontolyticus* (5 × 10⁵ CFU/mL), *R. dentocariosa* (10⁵ CFU/mL), or *R. mucilaginosa* (10⁵ CFU/mL) and incubated overnight at 37 °C. In addition, THP cells were stimulated with each of the four strains of nitrate-reducing bacteria in combination with *H. pylori*. 

#### 2.4.2. Cytokine Measurement in Stimulated Culture Supernatants

The expression levels of cytokines (tumor necrosis factor (TNF)-α and interleukin (IL)-8) in the culture supernatants were measured using an enzyme-linked immunosorbent assay. In this experiment, 96-well microtiter plates were coated with capture TNF-α and IL-8 antibodies (Invitrogen, Carlsbad, CA, USA) and then incubated overnight at 4 °C. The plates were blocked with phosphate-buffered saline (PBS) containing 10% skimmed milk for 1.5 h and then washed three times with PBS containing 0.05% Tween 20. After the culture supernatants were added, the plates were incubated for 1 h and washed again three times. They were added to TNF-α and IL-8 antibodies separately, incubated for 1 h, and then washed again three times. Subsequently, they were incubated with 400-fold diluted streptavidin (Proteintech Group, Inc., Rosemont, IL, USA) at room temperature for 30 min. Finally, a coloring reagent, 4 mg/mL O-phenylenediamine (Wako Pure Chemical Industries, Ltd., Osaka, Japan), was prepared in citrate buffer with 0.01% H_2_O_2_ and then added to the wells. After 15 min, the reaction was stopped using 6 N sulfuric acid. Absorbance was determined at 492 nm using a MULTISKAN FC microplate reader (Thermo Fisher Scientific K.K. Tokyo, Japan).

#### 2.4.3. Cytokine mRNA Transcription in Stimulated Cells

The mRNA transcription levels of cytokines (IL-2, IL-12, IL-4, and IL-10) were measured using RT-PCR. THP cells were homogenized using a QIAshredder (QIAGEN, Hulsterweg, The Nederland). mRNA was extracted using a NucleoSpin^®^RNA kit (Takara Bio Inc.) in accordance with the manufacturer’s protocol, and then the cDNA was synthesized using the Superscript ™ III First-Strand Synthesis System for RT-PCR (Invitrogen) in accordance with the manufacturer’s protocol. The PCR reaction solution was prepared using TaKaRa Ex Taq DNA Polymerase (Takara Bio Inc.) following the manufacturer’s protocol. Table 2 lists the sequences of the primers used. The PCR cycling program of the β-actin, IL-10, and IL-12 was as follows: denaturation at 94 °C for 5 min and 34 cycles of 94 °C for 10 s, 58 °C for 10 s, and 72 °C for 15 s. When other primers were used, only the annealing temperature was changed, and the other conditions were retained. Annealing of the other primers was performed at 56 °C for IL-2 and 60 °C for IL-4. The PCR products were electrophoresed on a 2% agarose gel and then stained with ethidium bromide. After staining, the gel was photographed using a GelScene GS-GU gel photography device (Astec).

### 2.5. Measurement of MAPK Activity

#### 2.5.1. Sample Preparation

Human gastric adenocarcinoma AGS cells (10⁶ cells/mL) were stimulated with *H. pylori* (10^7^ CFU/mL), *A. oris* (5 × 10^7^ CFU/mL), *A. odontolyticus* (5 × 10^7^ CFU/mL), *R. dentocariosa* (10^7^ CFU/mL), or *R. mucilaginosa* (10^7^ CFU/mL) and then incubated for 3 h at 37 °C. The stimulated cells were centrifuged at 5000 rpm for 5 min to separate the supernatant from the cells. The cells were added with 50 µL of purified water and 50 µL of sodium dodecyl sulfate sample buffer and then heated at 100 °C for 10 min. 

#### 2.5.2. Western Blot

The prepared samples (10 µL) were added to each lane of the gel, electrophoresed at 90 V for 2 h, and then transferred onto polyvinylidene fluoride (PVDF) membranes at 100 mA for 4 h. The PVDF membranes were blocked overnight with PBS containing 10% skim milk, washed three times with PBS, incubated with primary antibodies, and then shaken for 2 h. The primary antibodies were anti-p38 (1:1000 dilution; Cell Signaling Technology, Danvers, MA, USA), anti-p-p38 (1:1000 dilution; Cell Signaling Technology), anti-ERK1/2 (1:1000 dilution; Cell Signaling Technology), and anti-p-ERK1/2 (1:2000 dilution; Cell Signaling Technology). After washing, the cells were added with a horseradish peroxidase-labeled anti-rabbit IgG secondary antibody (Cell Signaling Technology) diluted 2000-fold with PBS containing 10% skimmed milk and then shaken for 1 h. After washing, bands were developed using a Pierce^®^ Western blotting substrate kit (Thermo Fisher Scientific) and detected by chemiluminescence using an Amersham Imager 600 (GE Healthcare Life Sciences, Tokyo, Japan), and then the amount of luminescence was quantified. 

### 2.6. Cell Cycle Measurement

#### 2.6.1. Sample Preparation

AGS cells (10⁶ cells/mL) were incubated in fetal calf serum (FCS)-free F-12K medium at 37 °C for 24 h and then synchronized. The medium was replaced with F-12K medium containing FCS, and then the AGS cells were incubated with the culture supernatant of each of the four types of nitrate-reducing bacteria and *H. pylori* at 37 °C for 24 h. Trypsin was added to the culture plate, from which the F-12K medium was removed, and then the plate was incubated at 37 °C for 10 min. AGS cells were collected in 15 mL tubes, fixed with cold 100% methanol at 4 °C for 2 h, centrifuged at 1000 rpm for 10 min, and then decanted. After washing with PBS, the cells were centrifuged again at 1000 rpm for 10 min and then decanted. This process was repeated twice. The cells were incubated with 0.25 mg/mL RNase solution (Sigma-Aldrich, St. Louis, MO, USA) at 37 °C for 30 min. The cells were incubated overnight with propidium iodide solution (Sigma-Aldrich) at 4 °C.

#### 2.6.2. Measurement and Analysis of Cell Cycle Progression

Cell cycle progression (G1, S, and G2/M phases) was analyzed using a MACSQuant Analyzer (Miltenyi Biotec, Tokyo, Japan) equipped with MACSQuantify analysis software (Miltenyi Biotec).

### 2.7. Statistical Analysis

Experiments of the cytokine (TNF-α, IL-8), MAPK activity, and cell cycle were independently performed for three samples, and those results were expressed as mean and standard error (SE). Statistical analyses of the cytokine (TNF-α, IL-8), MAPK activity, and cell cycle measurements were performed using *t*-tests. Statistical significance was considered at *p* < 0.05.

## 3. Results

### 3.1. Cytokine Measurement in Stimulated Culture Supernatants

#### 3.1.1. *Actinomyces* Genus

TNF-α production was significantly lower in the THP cells stimulated with *A. oris* or *A. odontolyticus* alone than in those stimulated with *H. pylori* alone (*p* < 0.01; Figure 1A). Compared with *H. pylori* monoculture, the coculture of *A. oris* or *A. odontolyticus* with *H. pylori* significantly increased TNF-α production in the cells (*p* < 0.01). Similarly, IL-8 production was significantly lower in the THP cells stimulated with *A. oris* or *A. odontolyticus* alone than in those stimulated with *H. pylori* alone (*p* < 0.05 and *p* < 0.01, respectively; Figure 1B). Compared with *H. pylori* monoculture, the coculture of *A. oris* or *A. odontolyticus* with *H. pylori* significantly increased IL-8 production in the cells (*p* < 0.01 and *p* < 0.05, respectively). 

#### 3.1.2. *Rothia* Genus

The amount of TNF-α produced in the THP cells stimulated with *R. dentocariosa* or *R. mucilaginosa* alone was comparable to that in the cells stimulated with *H. pylori* alone (Figure 1C). Compared with *H. pylori* monoculture, the coculture of *R. dentocariosa* or *R. mucilaginosa* with *H. pylori* significantly increased TNF-α production in the cells (*p* < 0.01 and *p* < 0.05, respectively). No significant difference in IL-8 production was found between the cells stimulated with *R. dentocariosa* or *H. pylori* alone (Figure 1D). However, the IL-8 production in the cells stimulated with *R. mucilaginosa* alone was significantly lower than that in the cells stimulated with *H. pylori* alone (*p* < 0.05). Compared with *H. pylori* monoculture, the coculture of *R. dentocariosa* or *R. mucilaginosa* with *H. pylori* significantly increased IL-8 production (*p* < 0.01).

### 3.2. Cytokine mRNA Transcription

#### 3.2.1. *Actinomyces* Genus

The THP cells stimulated with *A. oris*, *A. odontolyticus*, or *H. pylori* showed higher transcription levels of IL-2, IL-4, IL-12, and IL-10 than the control without stimulation (Figure 2A). No significant differences in the mRNA transcription levels of these cytokines were found between the THP cells stimulated with *A. oris* or *A. odontolyticus* alone and those stimulated with *A. oris* or *A. odontolyticus* and *H. pylori*. 

#### 3.2.2. *Rothia* Genus

The transcription levels of IL-2, IL-12, and IL-10 were higher in the THP cells stimulated with *R. dentocariosa*, *R. mucilaginosa*, or *H. pylori* than the control without stimulation (Figure 2B). No significant difference in IL-4 expression was found between the THP cells stimulated with *R. dentocariosa* or *R. mucilaginosa* and the control without stimulation. In addition, no difference in IL-12 expression was found between the THP cells stimulated with *R. dentocariosa* or *R. mucilaginosa* alone and those stimulated with *R. dentocariosa* or *R. mucilaginosa* and *H. pylori*. Meanwhile, the coculture of *R. dentocariosa* or *R. mucilaginosa* with *H. pylori* attenuated the expression of IL-2 and IL-10.

### 3.3. MAPK Activity

The expression of p-ERK/ERK in the cells stimulated with *A. oris*, *A. odontolyticus*, or *H. pylori* was significantly higher than that in the control (*p* < 0.01; Figure 3A). However, phosphorylation was weak in the cells stimulated with *R. dentocariosa* or *R. mucilaginosa*, and no significant difference was observed compared with the control. Similarly, p-p38/p38 was expressed in the cells stimulated with *A. oris*, *A. odontolyticus*, or *H. pylori*, but their expression levels were significantly higher in the cells stimulated with *A. oris* than the control (*p* < 0.05; Figure 3B). p-p38 was not expressed in the cells stimulated with *R. dentocariosa* or *R. mucilaginosa*.

### 3.4. Cell Cycle

The percentages of cells in the G1, S, and G2/M phases are shown in a pie chart (Figure 4). The percentage of cells in the G1 phase significantly decreased after stimulation with *A. oris*, *A. odontolyticus*, or *H. pylori* compared with the control (*p* < 0.05), confirming the transition from the G1 phase to the subsequent cycles. Only the cells stimulated with *H. pylori* showed a significantly increased rate of the S phase (*p* < 0.05). The number of cells in the G2/M phase increased after stimulation with *A. oris*, *A. odontolyticus*, or *H. pylori* (Figure 4A). No difference in cell cycle was observed in the cells stimulated with *R. dentocariosa* or *R. mucilaginosa* compared with the control (Figure 4B).

## 4. Discussion

*H. pylori* infection is an important risk factor for developing gastric cancer [5]. However, only a minority of *H. pylori*-infected individuals develop gastric cancer [6], suggesting that other risk factors are involved. The present study showed that nitrate-reducing bacteria isolated from the gastric mucosa of patients with atrophic gastritis caused by *H. pylori* infection might be risk factors for developing gastric cancer.

Chronic *H. pylori* infection activates neutrophils, lymphocytes, and macrophages in the host and induces the release of proinflammatory (e.g., IL-1β, IL-6, IL-8, TNF-α) and anti-inflammatory (e.g., IL-10) cytokines [18]. In the present study, the cocultivation of nitrate-reducing bacteria, *Actinomyces* and *Rothia,* with *H. pylori* increased TNF-α and IL-8 production, suggesting that coculture with *H. pylori* enhanced cytotoxicity and induced a strong acute inflammatory response. We also examined the expression of the cytokines involved in Th1/Th2 immune induction. During this specific immune response, IL-12 induces Th1 immunity to increase IL-2 production. In addition, IL-10 induces Th2 immunity and antibody production, consequently increasing IL-4 production [19]. In the present study, the *Actinomyces* and *Rothia* genera induced Th1 immunity similar to *H. pylori*, and no difference was observed between *Actinomyces* and *Rothia*. Meanwhile, *Actinomyces* can induce Th2 immunity similar to *H. pylori*. However, the inducing effect of *Rothia* on IL-4 was weak, and *Actinomyces* can induce Th2 immunity better than *Rothia*. This result suggests the presence of differences in Th2 immunity inducibility among the same nitrate-reducing bacteria. However, this hypothesis requires further investigation.

Activation of the MAPK pathway, a signaling pathway activated by *H. pylori* infection, affects cell cycle progression. ERK activation promotes G1 phase progression, and p38 activation induces the G2/M phase checkpoint [20,21]. *H. pylori* inhibits G_2_/M to G_1_ progression and gastric epithelial cell division [22]. The stimulation of cells with *H. pylori* and *Actinomyces* activates ERK and p38, resulting in a decreased G1 phase ratio and an increased G2/M phase ratio. In this study, *Actinomyces* spp. and *H. pylori* activated MAPK and affected the cell cycle, but not in Rothia spp. These results suggest that infection with these bacteria might affect cell cycle progression via the MAPK pathway in the stomach. However, there are limitations to this experiment because of the different growth times of those bacteria, and also, no long-term effects can be observed using this cell line. Other experimental systems would be necessary to prove the effect of carcinogenesis from epithelial cells.

## 5. Conclusions

Long-term *H. pylori* infection progresses to atrophic gastritis, and bacteria other than *H. pylori* proliferate and colonize the low-acid gastric mucosa. These symbiotic bacteria exacerbate the cytotoxicity and inflammation induced by *H. pylori* infection and affect the cell cycle, which may further advance carcinogenesis.

## Figures and Tables

**Figure 1 microorganisms-10-02495-f001:**
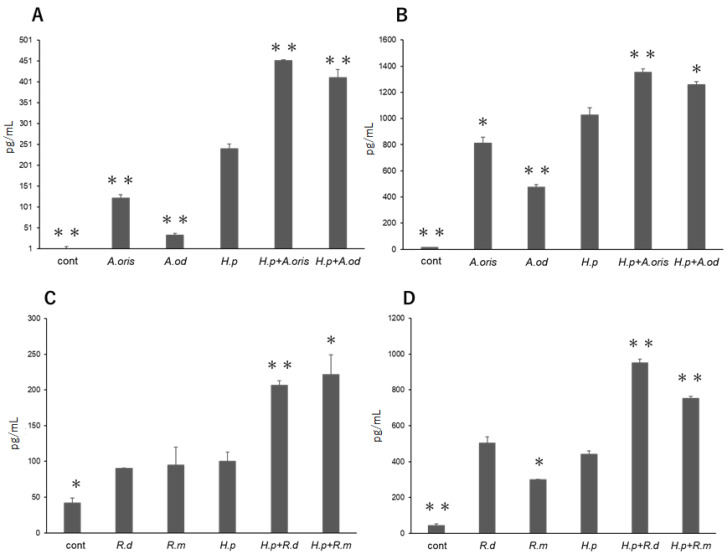
Cytokine production from THP cells stimulated with bacteria. TNF-α productions (*n* = 3, pg/mL) stimulated with *Actinomyces* genus and *H. pylori* (**A**), and with *Rothia* genus and *H. pylori* (**C**). IL-8 production (*n* = 3, pg/mL) by stimulated with *Actinomyces* genus and *H. pylori* (**B**), and with *Rothia* genus and *H. pylori* (**D**). (* *p* < 0.05, ** *p* < 0.01 vs. *H. pylori*).

**Figure 2 microorganisms-10-02495-f002:**
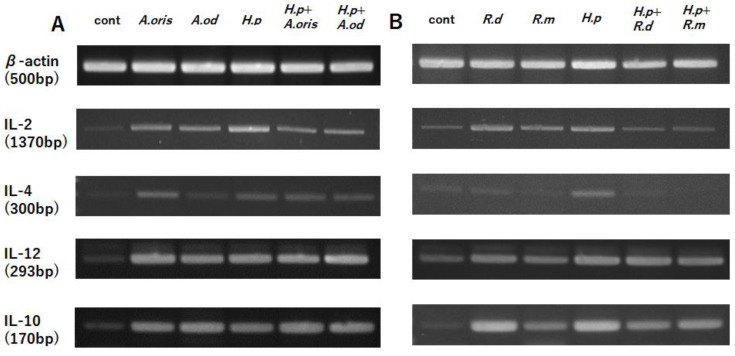
Transcription of cytokine mRNA. The mRNA transcription levels of cytokines (IL-2, IL-12, IL-4, and IL-10) were detected using RT-PCR and gel electrophoresis. (**A**) Stimulation with *Actinomyces* spp. and/or *H. pylori*; (**B**) stimulation with *Rothia* and/or *H. pylori*.

**Figure 3 microorganisms-10-02495-f003:**
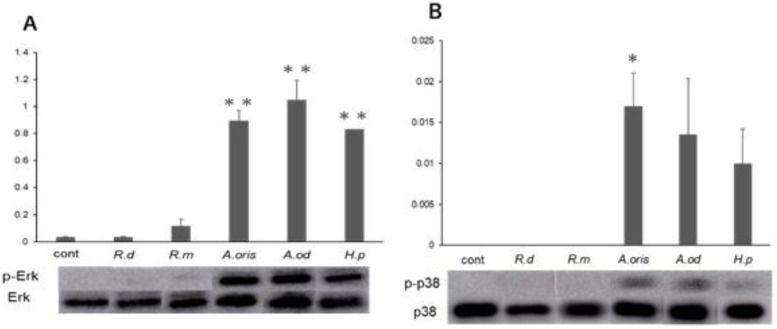
MAPK activity stimulated with each strain. Phosphorylation MAPK proteins were detected using Western blotting, and amounts of each protein expression (*n* = 3) were measured using an Amersham Imager 600. (**A**) p-ERK/ERK, (**B**) p-p38/p38 (* *p* < 0.05, ** *p* < 0.01 vs. control).

**Figure 4 microorganisms-10-02495-f004:**
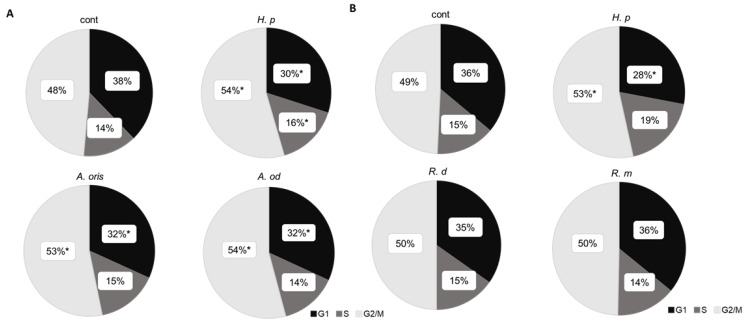
Cell cycle stimulated with bacterial supernatant. Gastric epithelial cells (AGS) were stimulated (*n* = 3) with each strain, and cell cycle phases were analyzed using MACSQuant analyzer. (* *p* < 0.05 vs. control). (**A**) Stimulation with *H. pylori* or *Actinomyces* spp.; (**B**) stimulation with *H. pylori* or *Rothia* spp.

**Table 1 microorganisms-10-02495-t001:** Isolated nitrate-reducing bacteria from stomach.

Species	Number of Strains
*Actinomyces oris*	2
*Actinomyces odontolyticus*	3
*Rothia mucilaginosa*	5
*Rothia dentocariosa*	3
*Streptococcus salivarius*	2
*Streptococcus vestibularis*	2
*Streptococcus oralis*	1
*Streptococcus sinensis*	1
*Staphylococcus aureus*	3
*Bacillus subtilis*	1
*Neisseria subflava*	1
*Klebsiella pneumoniae*	1
*Echerichia fergusonii*	1

**Table 2 microorganisms-10-02495-t002:** RT-PCR primers for cytokine.

Primer Name	Sequence	Tm (°C)
β-actinF	5′-GAGGCATCCTCACCCTGAAG-3′	58
β-actinR	5′-TCTTGGGATGGGGAGTCTGT-3′	
IL-2F	5′-CGTAATAGTTCTGGAACTAAAGGG-3′	56
IL-2R	5′-TGGGAAGCACTTAATTATCAAGTC-3′	
IL-12F	5′-ATTTAACGTTTGTCTGCCAGGATGT-3′	58
IL-12R	5′-ACATGGGAACTAGCATCTTGTTCTC-3′	
IL-4F	5′-AGCCTCACAGAGCAGAAGAC-3′	60
IL-4R	5′-CCGTTTCAGGAGTCGGATCA-3′	
IL-10F	5′-CGCTTCCCGAAGTAACAAGG-3′	58
IL-10R	5′-CACTGGGTAGCTTCTTTCGG-3′	

## Data Availability

The datasets generated and/or analyzed during the current study are available from the corresponding author upon reasonable request.

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
