# Peer review of "Effects of Helicobacter pylori and Nitrate-Reducing Bacteria Coculture on Cells"

_microorganisms, 2022, doi:10.3390/microorganisms10122495_

Round 1
Reviewer 1 Report
1. Should exclude the “)” included after the numbers in the authors’ list
2. On the topic 2.1. lines 66-69. The authors should clarify how many patients composed the sample, also if the patients signed the Term of Free and Informed Consent, and the number of the approbation in the Ethics Committee.
3. The author should explain the N of all analyses and if they were done in duplicate or triplicate.
4. Line 101. Explain the abbreviation OD.
5. All figures need to be reformatted to follow the template of the Microorganisms. If there are multiple panels, they should be listed in the legend as: (a) Description of what is contained in the first panel; (b) Description of what is contained in the second panel. Not using Uppercase letters. I suggest increasing the size and quality of the figure to improve visualization.
6. All figures and tables should be cited in the main text as Figure 1, Table 1, not as (Fig.2B) in lines 189,193, 199, 203, 216, 223, 246, 252,253.
7. Line 281 should be reviewed - instead of G2 and G1 should be G1/M to G1?
8. The discussion is objective and easy to understand. However, in the last paragraph of the discussion, I suggest to the authors explain more about how the MAPK pathway is affected by the association of the bacterial used in the study and the limitations of the study.
9. All references should be formatted to follow the Microorganisms Journal rule.
Author Response
Major revisions made after carefully considering the comments raised by the referee 1 are:
- Should exclude the “)” included after the numbers in the authors’ list
We have exclude the “)” in the authors’ name.
- On the topic 2.1. lines 66-69. The authors should clarify how many patients composed the sample, also if the patients signed the Term of Free and Informed Consent, and the number of the approbation in the Ethics Committee.
We isolated 26 strains of nitrate-reducing bacteria from 79 patients with gastritis. We have described patients’ number and approval number for ethical committee in 2.1. (from line 70 to line 72)
- The author should explain the N of all analyses and if they were done in duplicate or triplicate.
The experiment was performed triplicate samples independently, and the results are expressed as mean and SE. This were described in topic 2.7. (from line 186 to line 188) and we additionally described (N=3) in each figure legend.
- Line 101. Explain the abbreviation OD.
OD means optical density, So that we have corrected the words. (Line 104)
- All figures need to be reformatted to follow the template of the Microorganisms. If there are multiple panels, they should be listed in the legend as: (a) Description of what is contained in the first panel; (b) Description of what is contained in the second panel. Not using Uppercase letters. I suggest increasing the size and quality of the figure to improve visualization.
All figures and legends were corrected as Journal form
- All figures and tables should be cited in the main text as Figure 1, Table 1, not as (Fig.2B) in lines 189,193, 199, 203, 216, 223, 246, 252,253.
We have corrected all shortenings as Fig. to Figure.
- Line 281 should be reviewed - instead of G2 and G1 should be G1/M to G1?
 According to the references, they measured G2/M to G1 of cell cycle. We think this exposition is not correct.
- The discussion is objective and easy to understand. However, in the last paragraph of the discussion, I suggest to the authors explain more about how the MAPK pathway is affected by the association of the bacterial used in the study and the limitations of the study.
Limitation of this cell cycle experiment was described in the revision. (from line 288 to 293)
- All references should be formatted to follow the Microorganisms Journal rule.
We have corrected references according to Journal rule.
We hope that the revised version of our paper is now suitable for publication in Microorganisms and we look forward to hearing from you at your convenience.

Reviewer 2 Report
This is an exciting area of research to help explain why some individuals infected long-term with H. pylori progress to malignant changes while others do not. The potential for other risk factors in driving epithelial cell neoplasia has always been in the back of everyone's mind for those working in the field.
There are minor suggestions/critiques for the authors:
1) Why use undifferentiated human THP instead of differentiated one, as this could provide a better understanding of the type of cell (macrophage vs monocyte) in this interaction?
2) Human THP cells are not good producers of cytokines of the Th1/Th2 profile (IL-2, IL-4, IFN-gamma), which could explain some of the low mRNA transcription. Although decreased mRNA expression by coculture of Rothia and H. pylori is exciting.
3) The size for Figure 1 and Figure 4 needs to be increased. It is difficult to see the labeling corresponding to each group or co-cultures and the cell cycle phases.
4) Why not use the AGS gastric epithelial cells in all experiments? These cells interact with H. pylori and potentially other opportunistic bacteria, and their response to this infection may be very informative to the ensuing inflammatory response.
Author Response
Major revisions made after carefully considering the questions and comments raised by the referee 2 are:
1) Why use undifferentiated human THP instead of differentiated one, as this could provide a better understanding of the type of cell (macrophage vs monocyte) in this interaction?
THP cells turn into macrophage forms upon stimulation with mitogen such as a PMA, but at that time they secrete a large amount of cytokines, making it difficult to measure the effect of stimulation by bacteria. Experiments were performed using cells that had not yet differentiated into macrophages.
2) Human THP cells are not good producers of cytokines of the Th1/Th2 profile (IL-2, IL-4, IFN-gamma), which could explain some of the low mRNA transcription. Although decreased mRNA expression by co-culture of Rothia and H. pylori is exciting.
 Thank you for your important point. We also believe that experiments with THP cells have limitations. Ideally, it would be clearer to experiment with human peripheral blood, which contains IFN-g producing T cells. However, this experiment revealed that co-cultivation with Actinomyces may affect the immune system. Similar results have been obtained in other mouse infection experiments, so it is possible that this experiment is not just for cells. For these reasons, we hope that you will understand the results of this experiment and accept the description in the paper.
3) The size for Figure 1 and Figure 4 needs to be increased. It is difficult to see the labeling corresponding to each group or co-cultures and the cell cycle phases.
Figure size was expanded in the revision.
4) Why not use the AGS gastric epithelial cells in all experiments? These cells interact with H. pylori and potentially other opportunistic bacteria, and their response to this infection may be very informative to the ensuing inflammatory response.
In this paper, we examined the effects of bacteria other than H. pylori isolated from the stomach. We hypothesize that these have two effects, one on inflammation and the other on epithelial cell proliferation. Therefore, THP cells were used for the effects on inflammation, and AGS cells were used for the effects on epithelial cells. I've added them to the introduction to clarify those points. (from line 63 to line 65)
We hope that the revised version of our paper is now suitable for publication in Microorganisms and we look forward to hearing from you at your convenience.
